# Respiratory Diseases with High Occupational Fraction in Italy: Results from the Italian Hospital Discharge Registry (2010–2021)

**DOI:** 10.3390/healthcare12242565

**Published:** 2024-12-20

**Authors:** Pierpaolo Ferrante

**Affiliations:** Department of Occupational and Environmental Medicine, Epidemiology and Hygiene, Italian National Workers’ Compensation Authority (INAIL), Via Stefano Gradi 55, 00143 Rome, Italy; p.ferrante@inail.it; Tel.: +39-0654872733

**Keywords:** mesothelioma, pneumoconiosis, sinonasal cancer, hypersensitivity pneumonitis, occupational respiratory diseases

## Abstract

Objectives: Occupational respiratory diseases represent a major public health concern worldwide. This study analyses the hospitalization costs and characteristics of four major occupational respiratory diseases: malignant mesothelioma (MM), sinonasal cancer (SNC), pneumoconiosis (PN), and hypersensitivity pneumonitis (HP). The findings are situated within the context of Italy’s population trends and healthcare system, offering insights into the economic and clinical burden of these diseases. Study Design: This retrospective, population-based study examines Italian hospitalizations for MM, SNC, PN, and HP during the period 2010–2021. The primary outcomes were the number of hospitalizations, length of stay, and associated cost. Costs were derived from charges linked to diagnosis-related groups (version 24) and major diagnostic category coding systems. Results: Though the Italian population is rapidly aging, the annual number and rate of hospitalizations declined by 35% over the study period. SNC hospitalizations aligned with the overall trend, PN and MM experienced faster declines, whereas HP admissions remained steady. MM emerged as the most resource-intensive (EUR 25 million yearly, with 86% attributable to occupation), followed by PN (EUR 10 million, entirely occupational), SNC (EUR 5 million, with EUR 650,000 occupational), and HP (EUR 2 million, with EUR 370,000 occupational). All studied diseases had an average length of stay exceeding the national one. The SNC admissions were the shortest (6.5 days) and least expensive (EUR 3647). In contrast, MM, PN, and HP had a mean length of stay exceeding 10 days, with admission costs averaging EUR 4700 for MM and EUR 4000 for PN and HP. The median age was the highest for PN (78 years) and MM (71 years), while SNC and HP patients had a median age of approximately 65 years. Conclusions: Consistent with their anticipated benefits, Italian workplace health regulations over the last three decades, including the 1992 asbestos ban and D.lgs. 81/2008, are associated with significant reductions in the hospitalization burden and an increased median age at discharge for MM and PN. In contrast, fewer conclusions can be drawn for SNC and HP due to their lower occupational fractions (10–20%). This finding suggests adding an occupational exposure flag in hospital records for acknowledged occupational diseases to enhance surveillance. Finally, this study provides the first estimate of the occupational fraction of hospitalization costs for the studied diseases in Italy.

## 1. Introduction

Occupational diseases remain a significant global public health concern, often underestimated due to the lack of structured epidemiological surveillance systems. According to the International Labour Organization, global occupational deaths increased by over 12% from 2000 to 2019, with 2.6 million of the 2.93 million deaths attributed to work-related diseases [1]. Occupational respiratory diseases, though largely preventable, contribute significantly to this toll. Workers are particularly vulnerable to respiratory disorders due to their high exposure to potentially harmful substances. At a rate of 12–16 breaths per minute of 0.5 L each [2], industrial workers breathe 14,000–16,800 L of air during a typical 40 h workweek [3], often in environments contaminated with hazardous dust, fibres, or fumes [4]. Such exposure increases the risk of developing occupational respiratory diseases, including pneumoconiosis (PN), hypersensitivity pneumonitis (HP), malignant mesothelioma (MM), and sinonasal cancer (SNC). Although the epidemiological characteristics of these diseases are well-documented, their cumulative impact on hospital services remains underexplored. To address this gap, examining the characteristics of these diseases and evaluating the effectiveness of implemented policies is crucial for understanding their burden on healthcare systems.

HP is a rare disease, occurring when susceptible individuals mount an immune response to the cumulative inhalation of specific antigens from the environment. While it was historically classified based on the duration of symptoms as acute, sub-acute, or chronic, the latest guidelines recommend classifying patients into non-fibrotic and fibrotic categories [5]. Incidence rates in the USA and Denmark converge on 1.2–1.9 cases per 100,000 inhabitants, with a US mortality rate of 0.7 per million in 2016 [6,7]. The estimated occupational attributable fraction (OAF) range is 19–34% [8,9]

PN refers to a group of chronic interstitial lung diseases characterised by pulmonary inflammation and fibrosis. It is caused by the accumulation of mineral dust in the lungs, leading to several radiographic abnormalities, including nodules and opacities, with pleural plaques being the hallmark of asbestos exposure [10,11]. Specific exposure is linked to distinct forms of the disease, with the most prevalent types being asbestosis (from asbestos fibres), silicosis (from silica dust), and coal workers’ pneumoconiosis (from coal dust). Rarer forms include berylliosis (from beryllium dust), siderosis (from iron dust), and aluminosis (from aluminium dust). Despite being largely preventable, the global prevalence of PN increased by 66% from 1990 to 2017 [12,13]. The estimated OAF is essentially 100% [14].

MM is a rare but aggressive asbestos-related cancer, mainly developing in asbestos workers 20–60 years after the initial contact. It primarily affects the pleura (80–85%) or peritoneum (15–20%) and very rarely develops in the pericardium and tunica vaginalis testis (1–2%). Despite global efforts to ban asbestos, the World Health Organization (WHO) estimates that 125 million workers worldwide are still exposed to asbestos [15]. The impact of asbestos in developed countries is profound [16], with consequences continuing to emerge even decades after the bans were implemented [17]. The OAF in Italy was estimated at 86.6% [18].

SNC is a rare tumour, accounting for 1% of all malignancies, with a global annual incidence rate of 0.5–1 per 100,000 individuals [19,20,21]. It predominantly affects males over the age of 50. The main occupational risk factors are wood, leather, and textile dust, nickel and chromium compounds, and formaldehyde. The estimated OAF range is 11.8–33% [18,22].

Primary public health prevention policies for occupational respiratory diseases focus on reducing the disease incidence by minimizing and controlling hazardous substances in workplace air as well as mitigating the progression and complications of diseases in affected patients. Measuring the impact of these diseases on the healthcare system over time provides valuable insights for healthcare decision makers in evaluating public health policies. Additionally, understanding their economic burden can guide compensation providers in adjusting insurance premiums. This paper, part of a larger project analysing hospital discharge data from the Italian health system [23], aims to estimate the hospital costs associated with major occupational respiratory diseases.

## 2. Methods

### 2.1. Settings

Italy operates a public health system managed at the local level, where regional governments exercise significant autonomy in allocating health funds from the central administration to organise and oversee local healthcare services. They typically cover hospital charges based on Diagnosis-Related Group (DRG) and Major Diagnostic Categories (MDC) coding systems. The central administration retains control over coordination functions, including archiving regional hospitalization data in the National Hospital Discharge Registry. Within this registry, patient diagnoses are coded using the ninth version of the International Classification of Diseases Clinical Modification (ICD-9-CM). The central administration also defines the National Standard Hospital Charges (NSHC), which represent the maximum allowable cost per DRG (version 24) for acute care and per MDC for rehabilitation or long-term care. For each case type, whether acute or rehabilitation/long-term care, the Ministry of Health establishes a length of stay threshold (*thr*), beyond which a specific daily cost is applied.

### 2.2. Participants

This study included hospitalization episodes in Italy from 2010 to 2021 with a primary or secondary diagnosis of HP, PN, MM, or SNC.

### 2.3. Outcomes

The primary outcomes were the annual number of hospitalizations, the length of hospital stay and the associated costs. Secondary outcomes included the admission characteristics.

### 2.4. Data Sources/Measurement

We requested from the Ministry of Health 2010–2021 data on coal workers’ PN (ICD9-CM = 500), asbestosis (ICD9-CM = 501), silicosis (ICD9-CM = 502), PN due to other inorganic dust (ICD9-CM = 503), unspecified PN (ICD9-CM = 505), SNC (ICD9-CM = 160), MM (ICD9-CM = 163), peritoneum (ICD9-CM = 158.8–158.9), pericardium (ICD9-CM = 164.1), farmer’s lung (ICD9-CM = 495.0), bagassosis (ICD9-CM = 495.1), bird-fanciers’ lung (ICD9-CM = 495.2), suberosis (ICD9-CM = 495.3), maltworker’s lung (ICD9-CM = 495.4), mushroom-worker’s lung (ICD9-CM = 495.5), maple-bark-stripper’s lung (ICD9-CM = 495.6), ventilation pneumonitis (ICD9-CM = 495.7), other specified allergic alveolitis and pneumonitis (ICD9-CM = 495.8), and unspecified allergic alveolitis and pneumonitis (ICD9-CM = 495.9). The Ministry of Health provided a dataset including the number and length of stay of hospitalizations, aggregated by patient characteristic, treatment, and hospital-level variables. Specifically, aggregation variables were year, gender, age group (0 years; 1–4 years; 5–13 years; 14 years; 15–24 years; 25–44 years; 45–64 years; 65–74 years; 75+ years), region of patient’s residence, primary medical treatment (coded by ICD-9-CM), DRGs (with distinctions between medical and surgical types), care activity (including pregnancy-related, acute care, long-term care, and rehabilitation), hospitalization regimen (ordinary or day-care), and outcomes at discharge (deceased, transferred, or discharged to residence). For analytical purposes, we recombined the age classes in 4 groups (0–24, 25–44, 45–64, and 65+ years) and categorized the 20 regions into 5 macro-areas: North-West (Aosta Valley, Piedmont, Lombardy, and Liguria), North-East (Trentino-Alto Adige, Friuli-Venezia Giulia, Emilia-Romagna, and Veneto), Center (Tuscany, Marche, Umbria, and Lazio), South (Abruzzo, Basilicata, Molise, Apulia, Campania, and Calabria), and Major Islands (Sicily and Sardinia). Following the approach used in the Ministry of Health annual reports, we applied the National Standard Hospital Charges (NSHC), as defined by the Ministry’s decree of 12 October 2012, and *thr*s, as specified in the decree of 18 December 2008. Excluding admissions to day hospital with medical DRGs, NSHC for acute care are cumulatively expressed for the entire period up to *thr* time and as a daily charge for any period beyond *thr*. For rehabilitation, long-term care, and day hospital admissions in acute care settings with medical DRGs, NSHC are set on a daily basis, distinguishing between periods up to and exceeding *thr*. Data on causes of death were downloaded from the WHO site (https://www.who.int/data/data-collection-tools/who-mortality-database, accessed on15 January 2024). We downloaded the Italian population (Pop) by age and territorial division from the site of the Italian Institute of Statistics (ISTAT) (https://demo.istat.it/app/?l=it&a=2020&i=POS, accessed on15 January 2024). Statistics on the total number of hospitalizations in Italy—including the number of admissions, length of stay, and estimated costs—were obtained from Ministry of Health reports [24,25]. As the reported costs do not account for additional expenses incurred due to the pandemic, we added an estimate of the extra costs related to COVID-19 hospitalizations [26].

#### Statistical Analysis

Since the ICD-9-CM coding system does not contain a specific code for MM, we estimated its proportion among admissions by applying the proportion of mesotheliomas to pleural tumours by site as observed in mortality data. We examined the number of admissions (*n*) and the total days of hospital stay (*GG*), along with the admission rate on the population (n/POP) and the proportion of hospital stay (${GG}_{disease}/{GG}_{tot}$), expressed as basis points (bp). From *n* and *GG* we excluded values related to pregnancy-related admissions. Cost assessment was conducted in several steps using the aggregated dataset of admissions. For each record, based on NSHC and *thr* times, we associated the corresponding charges within (*ch*_1_) and beyond (*ch*_2_) *thr* and calculated the cumulative threshold (*T*) for grouped admissions as follows:*T* = *thr* × *n*


Costs of admission for acute care (*ACC*), except those for day hospitals with medical DRGs, were assessed using the following formula:*ACC* = *ch*_1_ × *n* + min{0,*GG*-*T*} × *ch*_2_

where min{0,*GG*-*T*} is the total number of days exceeding *thr*. For rehabilitation, long-term care admissions and acute care to day hospital with medical DRG (*RLC*) costs were calculated as follows:*RLC* = min{*GG,T*} × *ch*_1_ + min{0,*GG*-*T*} × *ch*_2_
where min{*GG*,*T*} represents the cumulative days for the entire group of admissions if it is less than the cumulative threshold; otherwise, it is the cumulative threshold. All costs were deflated by means of annual consumer price indexes provided by the ISTAT to allow comparisons by years. For the OAF of malignant neoplasms, we used values specific to the Italian population: 86.6% for MM and 18.1% for SNC [18]. For PN, we considered the OAF to be 100% [14] and for HP we used an OAF of 11.8%, as determined by a recent metanalysis [8].

## 3. Results

### 3.1. Italian Population

During the study period, the Italian population stood at about 60,000,000, showing an increasing trend until the peak of 60,345,917, reached in 2014, and followed by a subsequent decline. The overall median age was 45.3 years, with an increasing temporal trend during the study period (42.3 to 46.6).

### 3.2. Overall Hospitalizations

Overall hospital admissions were 105,910,190, averaging 8,825,849 per year and equivalent to a rate of 14.7 hospitalizations per 100 person-years. The annual number and rate of hospitalizations have been on a declining trend since 2010. Hospitalization days were 725,719,127, averaging 60,476,594 days per year, 1.0 days per person-year, and 6.9 days per admission. The average number of days per admission gradually increased from 6.5 in 2010 to 7.2 in 2021. The overall median age was 59.8 years, also increasing over time from 58.1 in 2013 to 61.0 in 2021. The overall annual expenditure for hospitalization care in Italy averaged EUR 29,688,895,583, representing 1.7% of the GDP in the entire period. This equates to EUR 3364 per admission and EUR 491 per hospital day. Yearly, while total costs decreased (EUR 34,267,904,922 to 27,616,154,755), costs per admission (EUR 3153 to 3933) and per day (EUR 485 to 543) increased (Table 1).

### 3.3. Overall Occupational Respiratory Disease Hospitalizations

From 2010 to 2021, there were 9625 yearly hospitalizations from the diseases studied, on average, totalling 93,650 hospital days (9.7 per admission) and accounting for a mean annual cost of EUR 41.6 million. In relative terms, there were 160.5 hospitalizations per 1,000,000 residents, representing 0.11% of total admissions, 0.16% of total hospitalization days, 0.14% of total hospitalization expenditures, and 0.0024% of gross domestic expenditures. The estimated parts of these figures attributable to occupational exposures are 7317 hospital admissions, 73,222 hospital days, and a cost of EUR 32.2 million. In relative terms, there are 122 hospitalizations per 1,000,000, representing 0.08% of total admissions, 0.12% of total hospitalization days, 0.11% of total hospitalization expenditures, and 0.0019% of gross domestic expenditures. The overall median age is 72.1 years, and males comprise 75.4% (Table 2).

### 3.4. Hypersensitivity Pneumonitis

On average, there were 473 admissions annually (7.9 hospitalizations per 10^6^ residents), corresponding to 4965 hospitalization days (0.8 bp of the total) and accounting for a cost of EUR 1.9 million. Until COVID-19, annual hospital admissions and costs remained stable. Afterward, both figures decreased to below 400 admissions and EUR 1,600,000, respectively. Similarly, annual hospitalization rates and proportions of hospital days had ranges of 7.1–9.9 and 0.7–0.9, respectively, before reaching the minimum during COVID-19. There were 90 admissions on average (ranging from 71 to 113) specifically due to occupational exposures, corresponding to 1.5 per 10^6^ residents (with a range of 1.2 to 1.9 per 10^6^ residents). These cases resulted in an average of 943 hospitalization days (ranging from 764 to 1218 days), accounting for 0.2% of all hospital days (ranging from 0.1 to 0.2%). The total cost associated with these occupational exposure-related hospitalizations was EUR 371,546, on average (ranging from EUR 301,345 to 454,677). The overall mean length of stay was 10.5 days, increasing from 9.9 in 2010 to 10.4 in 2021. The cost per admission was EUR 4116 (ranging from EUR 3807 to 4369), and the cost per day was EUR 394 (ranging from EUR 369 to 426), the percentage of males was 54.5%, the median age of patients was 65.4, and the percentage of surgical hospitalizations was 14.0% (Table 2 and Table 3).

### 3.5. Pneumoconiosis

On average, there were 2392 admissions (39.9 hospitalizations per 10^6^ residents) annually, corresponding to 24,388 hospitalizations days (4.0 bp of the total) and accounting for EUR 9,333,045. Annual hospital admissions (from 3963 to 994) and rates (66.4 to 16.8), hospital stay (39,425–9984), and its proportion against the total (5.6 to 2.0) as well as the total costs (EUR 15,001,573 to EUR 3,973,857) strongly decreased, particularly during the COVID-19 pandemic. All the reported admissions are attributable to occupational exposures. The overall mean length of stay was quite stable, at around 10.2 days. The overall cost per admission and per day were EUR 3902 and EUR 383, with annual figures showing an increasing trend from EUR 3785 to EUR 3998 and from EUR 381 to EUR 398, respectively. The percentage of males was 94.5%, the median age of patients was 77.9 years, and the percentage of surgical hospitalizations was 12.4% (Table 2 and Table 3).

### 3.6. Malignant Mesothelioma

On average, there were 5252 admissions (87.6 hospitalizations per 10^6^ residents) annually, corresponding to 52,772 hospitalization days (8.7 bp) and accounting for EUR 24,807,495. Annual hospital admissions (from 6329 to 3908) and rates (from 106.0 to 66.0), lengths of hospital stay (63,578 to 38,202), and related proportions against the total (9.0 to 7.5) as well as the total costs (29,132,190 to 19,546,472) strongly decreased over time. During the COVID-19 pandemic, the descent was faster. The part of these figures attributable to occupational exposures are 4548 admissions (ranging from 3384 to 5707), 45,701 hospitalization days (ranging from 33,083 to 55,576), and costs of EUR 21,483,292 (ranging from 16,927,245 to 25,228,477). The overall cost per admission and per day were EUR 4723 and EUR 470, both showing an increasing trend from EUR 4587 to EUR 5075 and from EUR 459 to EUR 511, respectively. The overall mean length of stay was quite stable, at around 10.0 days. The percentage of males was 71.1%, the median age of patients was 71.0 years, and the percentage of surgical hospitalizations was 30.4% (Table 2 and Table 3).

### 3.7. Sinonasal Cancer

On average, there were 1508 admissions (25.1 hospitalizations per 10^6^ residents) annually, corresponding to 11,525 hospitalization days (1.9 bp) for SNC and accounting for EUR 5,502,603. Annual hospital admissions (from 1857 to 1310) and rate (from 31.1 to 22.1), lengths of hospital stay (14,598 to 9464), and related proportion against the total (2.1 to 1.9) as well as the total costs (EUR 7,009,410 to EUR 4,734,643) decreased over time. The part of figures attributable to occupational exposures is 178 admissions (range of 153–219), 1360 hospitalization days (range of 1117–1723), and costs of EUR 649,307 (range of EUR 558,688–EUR 827,110). The overall costs per admission and per day were quite stable, at around EUR 3649 and EUR 477, respectively. The mean length of stay was quite stable, at around 7.6 days. The percentage of males was 65.5%, the median age of patients was 65.9 years, and the percentage of surgical hospitalizations was 46.2% (Table 2 and Table 3).

## 4. Discussion

Hospitalization trends play a crucial role in contextualizing occupational health policies within the broader healthcare framework, especially for diseases with long latency periods. This study offers a comprehensive high-level analysis of hospitalization patterns and associated economic burdens of four major occupational respiratory diseases in Italy: mesothelioma, pneumoconiosis, hypersensitivity pneumonitis, and sinonasal cancer. Though the occupational component of these diseases is preventable, they still impose a significant burden on the healthcare system, driven by past and potentially ongoing exposure, with a disproportionate impact stemming from their chronic and complex nature.

### 4.1. Underlying Trends

Over the past decade, the Italian population has aged notably, with a 10% increase in median age. This trend has inevitably led to a higher individual demand for healthcare services, as evidenced by the rise in the average length of hospital stays (from 6.5 to 7.3 days) and costs per admission (from EUR 3200 to EUR 3900). Conversely, healthcare cost-saving measures have resulted in a 25% reduction of hospitalization-cost-to-GDP ratio, leading to an overall decline of hospitalizations and expenditures by 35 and 28%, respectively. During the first years of the COVID-19 pandemic (2020–2021), hospitals were overwhelmed by COVID-19 cases, which significantly reduced hospitalizations for other conditions, including the studied diseases.

### 4.2. Occupational Respiratory Diseases

With the percentage of males among MM and PN hospitalizations of 71.5 and 94.4% respectively, gender data support the assumption that these diseases are both strongly tied to occupational exposures (with occupational fractions estimated at 86 and 100%, respectively). Although both disorders have experienced reductions in hospital admission numbers higher than the overall trend, they still represent a significant portion of hospital resources (0.11% for those with any ethology, and 0.10% for those with occupational ethology). The strong decline in hospitalizations is partially compensated by hospital stays longer (exceeding 10 days for MM, PN, and HP and 7.6 days for SNC) than the average (6.9 days), often involving older patients with multiple health conditions consuming more resources.

Hypersensitivity pneumonitis and sinonasal cancer, which have a lesser fraction of occupational aetiology (19 and 11%, respectively), showed declines in hospital admissions that were lower and in line with the overall trend, respectively. Further investigation is warranted to understand these patterns, as they could be attributed to several factors: (1) the reductions in occupational health disorder fractions may not have been significant within the whole set of diseases; (2) the reductions in occupational health disorders may have been compensated by growing awareness and detection rates; and (3) there may be gaps in occupational health protections.

As expected, given their localized but progressive/aggressive nature, surgery was used more frequently to treat tumours (46.2% for SNC and 30.6% for MM) compared to interstitial diseases (13.9% for HP and 12.4% for PN). Furthermore, the highest percentage of surgical cases in SNC was associated with the shortest hospital stays. Advancements in endoscopic techniques have likely contributed to the increased frequency of surgical treatments in SNC, as these less invasive approaches improve patient outcomes and reduce recovery times [27].

### 4.3. Implications and Recommendations

Work-related interventions over the past three decades, such as the 1992 asbestos ban, restrictions on silica use in high-risk industrial activities (including dry-sandblasting operations on ships) [28], and the establishment of the consolidated act on occupational health and safety in 2008 (D.lgs. 81/2008) have been associated with significant declines in the aggregated hospital burden and the highest median age at discharge of MM and PN hospitalizations during 2010–2021. These findings are consistent with the expected benefits of policies aimed at reducing occupational exposures, although other factors may also have played a role. Fewer conclusions can be drawn for SNC and HP due to their lower occupational fractions (10–20%). Incorporating an occupational exposure flag in hospital records to recognize occupational diseases is suggested to enhance surveillance efforts. It is also worth noting that in the Italian version of the ICD-9-CM coding system, the label associated with the code “504” (including byssinosis, Cannabinosis, and Flax Carder’s Disease) has been translated as “Pneumoconiosi da inalazione di altre polveri” instead of “Pneumopatia da inalazione di altre polveri”. While this error does not affect the study’s analysis, it should be corrected promptly to ensure data consistency. Additionally, current gaps in early detection in the studied diseases could be addressed by integrating all available datasets with the aid of new technologies. For instance, future research could focus on developing multi-biomarker predictive models powered by machine learning, using routine blood tests and general health parameter screenings. Finally, the estimates of total and occupational fractions of hospital costs can guide insurance compensation providers and inform adjustments to insurance premiums.

### 4.4. Study Limitations

This study has two main limitations. Firstly, patient diagnoses are coded using ICD-9-CM codes, which does not include specific codes for MMs. As a result, adjustments using mortality data coded by ICD-10 were necessary to estimate the fractions of MM among tumours by site. Secondly, we included all patients with at least one of the studied diagnoses as reported in the hospital discharge data. An overestimation of specific disease costs may have occurred, as up to six diseases can be recorded for each hospitalization. Nevertheless, given that these conditions are predominantly chronic, their hospitalization costs provide valuable insights into the overall health profile of affected individuals.

## 5. Conclusions

The results of this study align with the expected benefits of past data-driven policies aimed at reducing occupational exposure to asbestos and silica dust in Italy, while highlighting the potential for leveraging current technological advancements to integrate and link more comprehensive data. They also suggest adding an occupational exposure flag in hospital discharge data to allow refinements in occupational epidemiology. Finally, the hospital cost estimates provided enhance the knowledge base for insurance premium calculations.

## Figures and Tables

**Table 1 healthcare-12-02565-t001:** Italian population and hospitalizations per year (2010–2021).

Year	Population	Median Age (Years)	Hospitalizations					
			*n*	Median Age (Years)	Cost/GDP (%)	Days per Admission	Cost per Admission (2021, EUR)	Cost per Day (2021, EUR)
2010	59,690,316	42.3	10,869,148	-	2.0	6.5	3153	485
2011	59,948,497	42.7	10,347,388	-	1.9	6.6	3162	480
2012	60,105,185	43.0	9,851,527	-	1.8	6.6	3142	473
2013	60,277,309	43.4	9,450,543	58.1	1.8	6.7	3190	478
2014	60,345,917	43.7	9,140,116	58.5	1.8	6.8	3261	481
2015	60,295,497	44.2	8,930,979	59.2	1.7	6.9	3335	485
2016	60,163,712	44.6	8,697,574	59.6	1.7	6.9	3401	492
2017	60,066,734	45.0	8,522,456	60.1	1.6	6.9	3387	490
2018	59,937,769	45.4	8,357,575	60.5	1.6	7.0	3441	491
2019	59,816,673	45.8	8,212,494	60.9	1.6	7.0	3461	491
2020	59,641,488	46.2	6,508,840	61.7	1.5	7.5	3896	520
2021	59,236,213	46.6	7,021,550	61.0	1.5	7.2	3933	543
Mean	59,960,443	45.3	8,825,849	59.8	1.7	6.9	3364	491
Var ^1^	−0.8	10.2	−35.4	5.0	−25.0	11.6	24.7	12.0

^1^ Var = 2010 to 2021 relative variation (%).

**Table 2 healthcare-12-02565-t002:** Hospitalization characteristics of respiratory diseases.

Variable	Hypersensitivity Pneumonitis		Mesothelioma		Pneumoconiosis		Sinonasal Cancer		All Diseases	
	All	Occ ^1^ (19%)	All	Occ ^1^ (86.6%)	All	Occ ^1^ (100%)	All	Occ ^1^ (11.8%)	All	Occ ^1^
Hospitalizations										
*per year*	473	90	5252	4548	2392	2392	1508	178	9625	7208
*proportion (bp* ^2^ *)*	0.5	0.1	5.9	5.2	2.7	0.5	1.7	0.2	10.9	8.2
*rate* ^3^ × 10^6^	7.9	1.5	87.6	75.9	39.9	39.9	25.1	3.0	160.5	120.2
Length of stay										
*annual days*	4965	943	52,772	45,701	24,388	24,388	11,525	1360	93,650	72,392
*proportion (bp* ^2^ *)*	0.8	0.2	8.7	7.6	4.0	4.0	1.9	0.2	15.5	12.0
*per admission*	10.5		10.0		10.2		7.6		9.7	10
Cost										
*per year*	1,955,506	371,546	24,807,495	21,483,291	9,333,045	9,333,045	5,502,603	649,307	41,598,649	31,837,189
*per admission*	4134		4723		3902		3649		4322	4417
*per day*	394		470		383		477		444	440
Cost relative to										
*all admission (bp* ^2^ *)*	0.7	0.1	8.4	1.6	3.1	0.6	1.9	0.2	14.0	10.7
*GDP (bp* ^2^ *)*	<0.1	<0.1	0.2	0.2	0.1	0.1	<0.1	<0.1	0.2	0.2
Median age	65.4		71		77.9		65.9		72.1	
Male (%)	54.5		71.1		94.5		65.5		75.4	
Surgery (%)	14.0		30.4		12.4		46.2		27.7	

^1^ Occ = due to occupational exposure. Parentheses contain the fraction attributable to occupation used for the analysis. ^2^ bp = basis points (‱ or ×10^−4^). ^3^ rate = No. admissions/population.

**Table 3 healthcare-12-02565-t003:** Hospitalization trends of hypersensitive pneumonitis (HP), malignant mesothelioma (MM), pneumoconiosis (PN), and sinonasal cancer (SNC).

Disease	Year	Hospitalizations		Length of Stay			Median Age (Years)	Cost		
		*n*	Rate ^1^ × 10^6^	Total	Fraction (bp ^2^)	Per Adm ^3^		Total	Per Adm ^3^	Per Day
HP	2010	490	8.2	4866	0.7	9.9	61.5	2,072,286	4229	426
HP	2011	496	8.3	4914	0.7	9.9	63.9	2,011,476	4055	409
HP	2012	469	7.8	4795	0.7	10.2	62.3	1,882,274	4013	393
HP	2013	425	7.1	4356	0.7	10.2	62.9	1,617,933	3807	371
HP	2014	440	7.3	4457	0.7	10.1	63.9	1,853,149	4212	416
HP	2015	514	8.5	5428	0.9	10.6	65.6	2,002,274	3895	369
HP	2016	484	8.0	5231	0.9	10.8	65.7	1,983,646	4098	379
HP	2017	518	8.6	5422	0.9	10.5	66.2	2,251,449	4346	415
HP	2018	505	8.4	5534	0.9	11.0	67.4	2,206,359	4369	399
HP	2019	593	9.9	6410	1.1	10.8	66.4	2,393,039	4035	373
HP	2020	374	6.3	4144	0.8	11.1	64.0	1,606,166	4295	388
HP	2021	388	6.6	4020	0.8	10.4	66.0	1,586,025	4088	395
PN	2010	3.963	66.4	39,425	5.6	9.9	77.6	15,001,573	3785	381
PN	2011	3.622	60.4	35,716	5.2	9.9	77.8	14,201,263	3921	398
PN	2012	3.319	55.2	33,135	5.1	10.0	77.9	12,056,587	3633	364
PN	2013	2.961	49.1	30,933	4.9	10.4	77.8	11,367,942	3839	368
PN	2014	2.794	46.3	28,100	4.5	10.1	78.0	10,695,345	3828	381
PN	2015	2.309	38.3	24,490	4.0	10.6	78.1	9,468,695	4101	387
PN	2016	2.169	36.1	22,313	3.7	10.3	78.1	8,553,620	3944	383
PN	2017	2.077	34.6	22,094	3.8	10.6	78.3	8,505,529	4095	385
PN	2018	1.799	30.0	19,015	3.2	10.6	77.9	7,204,355	4005	379
PN	2019	1.609	26.9	16,328	2.8	10.1	78.0	6,489,207	4033	397
PN	2020	1.085	18.2	11,121	2.3	10.2	78.0	4,478,555	4128	403
PN	2021	994	16.8	9984	2.0	10.0	77.8	3,973,857	3998	398
MM	2010	6.329	106	63,578	9.0	10.0	69.0	29,132,190	4587	459
MM	2011	6.590	109.9	64,175	9.4	9.7	69.3	28,978,774	4461	451
MM	2012	6.157	102.4	61,841	9.4	10.0	70.1	27,702,556	4609	449
MM	2013	5.619	93.2	59,731	9.5	10.6	70.4	27,333,903	4756	458
MM	2014	5.443	90.2	55,678	9.0	10.2	70.8	25,579,782	4657	460
MM	2015	5.301	87.9	54,294	8.8	10.2	71.0	25,095,160	4771	463
MM	2016	5.295	88.0	53,043	8.8	10.0	71.8	24,614,943	4761	464
MM	2017	4.789	79.7	48,123	8.2	10.0	72.4	22,912,241	4776	477
MM	2018	5.016	83.7	50,874	8.7	10.1	72.5	24,278,780	4807	477
MM	2019	4.546	76.0	44,761	7.7	9.8	72.7	21,887,366	4908	488
MM	2020	4.031	67.6	38,964	8.0	9.7	72.3	20,627,796	5169	529
MM	2021	3.908	66.0	38,202	7.5	9.8	72.8	19,546,472	5075	511
SNC	2010	1.857	31.1	14,598	2.1	7.9	65.4	7,009,410	3775	480
SNC	2011	1.737	29.0	14,324	2.1	8.2	66.0	6,510,328	3748	455
SNC	2012	1.603	26.7	11,941	1.8	7.4	66.1	5,442,921	3395	456
SNC	2013	1.430	23.7	10,065	1.6	7.0	66.1	4,833,350	3380	480
SNC	2014	1.524	25.3	12,338	2.0	8.1	65.8	5,491,217	3603	445
SNC	2015	1.559	25.9	11,507	1.9	7.4	66.2	5,468,481	3508	475
SNC	2016	1.395	23.2	10,789	1.8	7.7	66.6	5,037,640	3611	467
SNC	2017	1.517	25.3	11,243	1.9	7.4	65.6	5,675,976	3742	505
SNC	2018	1.405	23.4	11,193	1.9	8.0	65.8	5,294,924	3769	473
SNC	2019	1.466	24.5	11,108	1.9	7.6	65.9	5,427,295	3702	489
SNC	2020	1.297	21.7	9721	2.0	7.5	66.7	5,105,036	3936	525
SNC	2021	1.310	22.1	9464	1.9	7.2	63.5	4,734,643	3614	500

^1^ rate = No. admissions/population. ^2^ bp = basis points (‱ or ×10^−4^). ^3^ Adm = Admission.

## Data Availability

Data are available on reasonable request. Data are released by the Ministry of Health for research purposes.

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
