# Peer review of "Respiratory Diseases with High Occupational Fraction in Italy: Results from the Italian Hospital Discharge Registry (2010–2021)"

_healthcare, 2024, doi:10.3390/healthcare12242565_

Round 1

Reviewer 1 Report (Previous Reviewer 2)

Comments and Suggestions for Authors

 General

The concerns raised by the reviewer have been considered by the author and the revised manuscript has improved. There are still some issues to consider.

Major:

1.      Abstract, lines 42-45 and 49-51; Discussion, lines 503-510; Conclusions, lines 535-538: the study does not provide any data to support the statement that the reduction of hospitalization for occupational respiratory diseases is due to the past policies to reduce occupational exposures. It should be modified.

Minor:

-          Abstract there is no consistency in the abbreviation of sinonasal cancer (SC or SNC?)

-          Table 2: the headings of columns should be in English (Alveolite, Mesotelioma, ..)

-          Line 353: ‘fully preventable’ is probably too strong. Please,  consider ‘Despite the occupational component of these diseases is preventable …’

-          Line 505: It is not clear what ‘[Asbestosis 2021]’ is referred to.

Author Response

Introduction: The concerns raised by the reviewer have been considered by the author and the revised manuscript has improved. There are still some issues to consider.

Thank you for your kind words. Your comments provided valuable input and stimuli that greatly helped me improve my paper. Below, I outline the changes made to address your new concerns.

Comment 1: Abstract, lines 42-45 and 49-51; Discussion, lines 503-510; Conclusions, lines 535-538: the study does not provide any data to support the statement that the reduction of hospitalization for occupational respiratory diseases is due to the past policies to reduce occupational exposures. It should be modified.

Response: Thank you for pointing this out. I acknowledge that the observed association supports consistency rather than establishing a causal effect. Accordingly, I have revised the sentences highlighted in your comment. Additionally, I took the opportunity to emphasize the importance of estimating the occupational fraction of hospitalizations costs related to the studied diseases (lines 49-51 and 521-522).

  • lines 42-45:

“Consistent with their anticipated benefits, Italian workplace health regulations over the last three decades, including the 1992 asbestos ban and the D.lgs. 81/2008, have been associated with significant reductions in the hospitalization burden and an increased median age at discharge for MM and PN”

  • lines 49-51:
    “This finding suggests adding an occupational exposure flag in hospital records for acknowledged occupational diseases to enhance surveillance. Finally, this study provides the first estimate of the occupational fraction of hospitalization costs for the studied diseases in Italy”

  • lines 508-510:
  • “These findings are consistent with the expected benefits of policies aimed at reducing occupational exposures, although other factors may also have played a role”

  • lines 521-522:

“Finally, the estimates of total and occupational fractions of hospital costs can guide insurance compensation providers and inform adjustments to insurance premiums”

  • lines 535-538:

“The results of this study align with the expected benefits of past data-driven policies aimed at reducing occupational exposure to asbestos and silica dust in Italy, while highlighting the potential for leveraging current technological advancements to integrate and link more comprehensive data”

Comment 2: Abstract there is no consistency in the abbreviation of sinonasal cancer (SC or SNC?)

Response: Thank you for pointing this out. I replaced SC with SNC in lines 14,19

Comment 3:  Table 2: the headings of columns should be in English (Alveolite, Mesotelioma, ..)

Response: Thank you for pointing this out. I replaced Italian with English labels

Comment 4: Line 353: ‘fully preventable’ is probably too strong. Please,  consider ‘Despite the occupational component of these diseases is preventable …’

Response: Thank you for pointing this out. I replaced my sentence with the yours.

Comment 5: Line 505: It is not clear what ‘[Asbestosis 2021]’ is referred to.

Response: Thank you for pointing this out. I added the reference and cited the corresponding number Lines 503-505: “Work-related interventions over the past three decades, such as the 1992 asbestos ban, restrictions on silica use in high-risk industrial activities (including dry-sandblasting operations on ships) [28]”

reference

  1. Ferrante, P., 2021. Costs of asbestosis and silicosis hospitalization in Italy (2001–2018) Costs of asbestosis and silicosis hospitalization. International Archives of Occupational and Environmental Health, 94, pp.763-771.

Reviewer 2 Report (Previous Reviewer 1)

Comments and Suggestions for Authors

I thank the author for the work done on the paper

Author Response

Thank you for your time. Your comments provided valuable input and stimuli that greatly helped me improve my paper

Round 2

Reviewer 1 Report (Previous Reviewer 2)

Comments and Suggestions for Authors

 General

The concerns raised by the reviewer have been considered by the author and the revised manuscript has improved. There are still some issues to consider.

Major:

1.      Abstract, lines 42-45 and 49-51; Discussion, lines 503-510; Conclusions, lines 535-538: the study does not provide any data to support the statement that the reduction of hospitalization for occupational respiratory diseases is due to the past policies to reduce occupational exposures. It should be modified.

Minor:

-          Abstract: there is no consistency in the abbreviation of sinonasal cancer (SC or SNC?)

-          Table 2: the heading of columns should be in English (Alveolite, Mesotelioma, ..)

-          Line 353: ‘fully preventable’ is probably too strong. Please,  consider ‘Despite the occupational component of these diseases is preventable …’

-          Line 505: It is not clear what ‘[Asbestosis 2021]’ is referred to.

Round 3

Reviewer 1 Report (Previous Reviewer 2)

Comments and Suggestions for Authors

The second revision of the above manuscript is fine.

I suggest to accept the paper.

Author Response

This manuscript is a resubmission of an earlier submission. The following is a list of the peer review reports and author responses from that submission.

Round 1

Reviewer 1 Report

Comments and Suggestions for Authors

In this paper, Pierpaolo Ferrante and PhD (sic.) presented epidemiological data regarding hospitalisation time and work-related pulmonary disease costs. The author presented a broad evaluation of the Italian population alongside the single diagnosis trends over the last 12 years. The diagnoses evaluated are malignant pleural mesothelioma, hypersensitivity pneumonitis, sinonasal carcinoma and pneumoconiosis. Data from a centralized ICD-9-based dataset.

I want to address some points regarding this overall interesting paper, however, in my opinion, it is not focused enough on the topic presented in the introduction and the title:

Major points:

1)     My main point is regarding the focus of the paper; data regarding the epidemiology of these diseases are interesting, however, I believe the author reached conclusions out of its data and scope:

a)      First of all, the focus given to the general population and general hospitalization is too much for a paper based on the premise of evaluating 4 specific diseases. The paragraph on general population in the discussion is too extensive and out of scope;

b)     The paragraph in the discussion on COVID-19 is out of scope and only partially based on the data presented;

c)      All the evaluations on the endoscopic treatment of SNC is speculative and not based on the data presented by the author (there is no mention of evaluating the surgical approach of the patients), therefore conclusions based on those speculations cannot be the conclusions of this paper;

d)     Policy recommendations, even if based on common sense, are not based on the data presented by the paper, therefore I believe all the paragraph do not find space in an original paper, while it could belong in a literature review on this topic;

e)     I did not understand why the author cites artificial intelligence in its conclusions;

2)     In statistical analysis, it is not clear how the author reached the number of mesotheliomas: it seems he used the number of deaths associated with MM from single sites, however, this method can both underestimate or overestimate the number of patients (patients who died in other institutions or at home and not in the diagnostic center)

3)     Figure 1 is hardly understandable, does not have a description, the ordinate values are not readable, and the graphical presentation of only general population data is out of the scope of the paper in my opinion

Minor points:

-        The conclusions of the abstract seem from another paper based on the results presented in the same abstract, however, they are based on the conclusions of the paper in which I disagree. (see point 1 of major points)

-        Recent evolution in HP literature tended to avoid the division into acute and chronic disease, changing to a more radiological evaluation of non-fibrosing and fibrosing HP (DOI: 10.1183/16000617.0169-2021), so I suggest to implement this concept in the introduction. 

-          I do not understand by the author list if Dr. Ferrante worked with a fellow PhD or if he worked alone and is a PhD.

Comments on the Quality of English Language

The author must perform a typo check

Reviewer 2 Report

Comments and Suggestions for Authors

 General

The author performed a descriptive retrospective analysis of hospital admissions and estimated costs of occupational respiratory diseases in Italy between 2010 and 2021. The study provides some interesting information. However, the importance of the investigation is limited by the descriptive nature and the lack of comparisons with other periods of time and different countries.

There are several issues to be considered.

Major:

1.      Introduction, pag. 2, lines 59-68: byssinosis does not meet the definition of pneumoconiosis. It is due to inhalation of organic dust (endotoxins) without interstitial fibrosis. It should be deleted.

2.      Hospital Discharge Records in Italy (SDO in Italian) report several diagnoses (principal and secondary). It should be specified which diagnoses were considered in the analysis. Using principal diagnosis only may underestimate the occupational nature of the admission (i.e. a patient with silicosis admitted with acute respiratory failure will probably have the latter as principal diagnosis and silicosis as a secondary). Using all the diagnosis may cause a bias of information (again i.e. a patient with silicosis admitted for a bone fracture will have silicosis among the secondary diagnoses). Please, explain how the issue was managed.

3.      Figure 1: It is hardly readable (to small characters, pale green over white background). In addition, the panel were not identified. Finally (line 207), Figure 1d is mentioned but apparently Figure 1 has only 3 panels.

4.      Results: The text is redundant. The data shown already in Tables and Figure should be omitted.

5.      Discussion, para 5.4: It is not necessary since it reports obvious general considerations unrelated with the object of the study and the major findings.

6.      Conclusions (and abstract, lines 27-32): They are not in line with the findings of the investigation.

Minor:

-          There are several typos: line 21, ‘medin’; line 23. ‘od’; Table 1: ‘anno’

-          A legend for Table S1 should be provided (Med, MedO)

Comments on the Quality of English Language

Please, see above and check for typos
